# Safeguarding DNA Replication: A Golden Touch of MiDAS and Other Mechanisms

**DOI:** 10.3390/ijms231911331

**Published:** 2022-09-26

**Authors:** Baraah Al Ahmad Nachar, Filippo Rosselli

**Affiliations:** CNRS UMR9019, Gustave Roussy Cancer Campus, Université Paris-Saclay, Equipe Labellisée Ligue Nationale Contre le Cancer, 94801 Villejuif, France

**Keywords:** DNA replication, MiDAS, chromosome instability, FANCD2, DNA repair

## Abstract

DNA replication is a tightly regulated fundamental process allowing the correct duplication and transfer of the genetic information from the parental cell to the progeny. It involves the coordinated assembly of several proteins and protein complexes resulting in replication fork licensing, firing and progression. However, the DNA replication pathway is strewn with hurdles that affect replication fork progression during S phase. As a result, cells have adapted several mechanisms ensuring replication completion before entry into mitosis and segregating chromosomes with minimal, if any, abnormalities. In this review, we describe the possible obstacles that a replication fork might encounter and how the cell manages to protect DNA replication from S to the next G1.

## 1. Introduction

The faithful and timely replication of the genome relies on well-orchestrated events that ensure DNA synthesis in coordination with cell cycle progression and the equal segregation of chromosomes into the daughter cells at the end of mitosis. However, even under physiological conditions, replication forks face several types of barriers that generate a replication stress delaying or impeding their progression, affecting the efficiency of genome duplication and fueling genome instability. Several intrinsic and extrinsic sources of replication stress are recognized, the major ones being the presence of non-B DNA secondary structures [1], the replication–transcription conflicts [2], the deregulations in replication origin licensing and firing [3,4,5,6,7,8], the alterations in the equilibrium of the nucleotide pools [9,10] and the presence of DNA damage [11].

A challenging question in the field is whether and how DNA replication is safely maintained under low, i.e., intrinsic, or mild replication stress conditions and in what time frame it is completed [12,13,14]. Notably, intrinsic replication stress in otherwise unperturbed conditions contribute to S-phase extension in cancer cells [15], which require dedicated DNA repair and replication rescue mechanisms to maintain fork progression, stability, and restart allowing the completion of genome replication before the onset of mitosis. Notably, under conditions of low or mild replication stress, immunofluorescence analysis of late G2 and mitotic cells previously shortly incubated (20–60 min) with the thymidine analogues ethyl-2′-deoxyuridine (EdU) or 5-bromo-2′-deoxyuridine (BrdU) revealed nucleotide incorporation as bright ‘dots’, consequence of the synthesis of DNA stretches far beyond the classical limits of S-phase [12,13,14]. This raised the question whether the observed dots represent delayed DNA replication process of ‘reparative’ DNA synthesis taking place at the end of the incomplete genome duplication.

In counteracting replication stress, the FANConi/BReast CAncer (FANC/BRCA) pathway, its regulators and targets represent the major elements of the DNA damage response (DDR) network dedicated to the safeguard of the replication fork ensuring the timely accurate duplication and segregation of the genome. More than 20 different *FANC* genes encoding the FANC/BRCA pathway proteins have been identified to date, including several genes previously identified in heterozygous carriers at risk of breast and ovarian cancer (BOC) and directly involved in homologous recombination (HR) [16,17,18,19,20]. The genetic inactivation of the FANC/BRCA pathway leads to Fanconi anemia (FA), a hematologic disorder associating aplastic anemia and predisposition to cancer with chromosomal fragility and belonging to the Inherited Bone Marrow Failure Syndromes [21,22]. The key biochemical event that turn on the FANC/BRCA pathway is the FANCcore complex-mediated monoubiquitination of FANCD2 and FANCI [23]. This reaction occurs during DNA replication and peaks in the presence of DNA damage. FANCD2 and FANCI mono-ubiquitination has been shown to be essential in helping cells to cope with replication stress [24,25]. Indeed, monoubiquitinated FANCD2 (monoUbFANCD2) and mono-UbFANCI assemble in subnuclear foci at the chromatin where, in a yet undetermined way, they coordinate the events necessary for DNA repair, if any, and replication rescue.

In this review, we will discuss how the cell controls genome replication during the canonical S-phase, how the FANC pathway contributes to the replication stress response ensuring faithful chromosome segregation, and the possible meaning of extra S-phase DNA synthesis.

## 2. DNA Replication and Replication Stress

### 2.1. Origin Licensing and Firing

A complete genome duplication depends on the assembly and licensing of a sufficient quantity of replication origins onto the DNA and the successive timely and progressive firing of a subset of such licensed replication origins throughout the entire S-phase [26,27,28,29]. To ensure the activation of a single origin only once throughout cell cycle, DNA replication proceeds in two exclusive steps. The first, which takes place from the end of mitosis until end of G1, is known as “origin licensing”. The second, the progressive activation of some of the licensed origins during S phase, is known as origin firing. This type of regulation is conserved throughout evolution emphasizing the importance of its role for cell survival. Origin licensing is characterized by the assembly of the six proteins of the Origin Recognition Complex (ORC1-6) [30], the Cell Division Cycle 6 (CDC6) [31], the CDC10-Dependent cell division cycle Transcripts 1 (CDT1) [32] together with the Mini Chromosome Maintenance (MCM2-7) proteins constituting the replicative helicase [33] to the chromatin in the presence of a low Cyclin-Dependent Kinase (CDK) activity. This protein complex is called pre-replication complex (pre-RC). Around 100,000 origins of replication are licensed per mammalian cell. However, in normal conditions, only 10% of them initiate and complete DNA replication during unperturbed S phase while the other 90% remain dormant [34,35]. These dormant origins have backup functions and can be triggered in the event of a fork stall or slowdown to rescue or maintain replication rate [34,35].

The progressive firing of such a subset of licensed origins starts at S-phase entry. CDK and DbF4-dependent CDC7 kinase (DDK) phosphorylates and loads MCM-10, DNA helicase Q4 (RECQL4), Cell Division Cycle protein 45 (CDC45), (Go, Ichi, Nii, and San; five, one, two, and three in Japanese) (GINS), Topoisomerase Binding Protein 1 (TOPBP1) and Treslin on the pre-RC to form the pre-initiation complex (pre-IC) [36]. CDC45, MCM, and GINS, the CMG complex, enhance the helicase activity of MCM2-7 allowing DNA double helix opening [37]. Upon activation, the double MCM hexamers dissociate into two hexamers to form with other proteins two active replisomes traveling in opposite directions. Each replisome is associated with two DNA replication machineries replicating the leading and lagging strands. Along with the activation of helicases, several other replication factors and complexes are recruited to the chromatin including the ssDNA-binding replication protein A (RPA), the processivity ring formed by the homo-trimeric assembly of the proliferating cell nuclear antigen (PCNA), the replication factor C (RFC) complex, and several DNA polymerases. RPA complex controls the loading of DNA polymerase α, which mediates short primers synthesis on both leading and lagging strands [38]. Successively, the RFC complex binds to the primers and loads PCNA to the chromatin allowing primers elongation [39] recruiting DNA Polymerase ε, which contributes to the elongation of leading strand, and DNA polymerase δ, which is involved in the elongation of Okazaki fragments [40] (see [41] for exhaustive review).

Deregulation in origins licensing or firing affects the rate of DNA synthesis and results in replication stress and chromosome instability [5,6,42,43].

### 2.2. Regulation of DNA Replication in Space and Time

Different DNA regions are replicated at different times during S phase [44] and at different places inside the nucleus (see Figure 1). Individual replication origins are arranged in a replicon or replication unit, a DNA region which, physiologically, will be replicated by a single fired origin despite the presence of other dormant origins (see Figure 1A) [45]. Replicons are arranged in clusters. Each “to be fired” origin inside a replication unit arranged in the same cluster fires at the same time through positive interference mechanism [46]. It has been suggested that cohesins play a role in tethering several replicons together to form a single replication domain or a replication factory (see Figure 1B) [47]. Groups of replication domains are fired at different time points through the S-phase, assuring the optimal intake of elements, including proteins, nucleotides and energy supply, necessary to the process avoiding their exhaustion. The progressive firing of timely separated replication domain, the so-called replication timing, is also under the control of the activity of CDK and DDK kinase as well as dependent on Treslin expression and phosphorylation [48].

Replication foci (≈ replication domains) are spatially and structurally differentially organized inside the nucleus (see Figure 1C). The early replicating domains correspond to the euchromatic regions of the nucleus, gene-rich and actively transcribed chromosome regions [49] and are organized as topologically-associated domains (TADS) inside the nucleus. In contrast, the late-replicating domains are heterochromatin, gene-poor regions associated to the nuclear lamina (LAD) [50]. Both early and late replicating domains are referred to as constant timing regions (CTR) and are separated by large DNA segments devoid of origins. These DNA segments are referred to as timing transition regions (TTR). Normally, TTR are the latest to replicate since they lack intrinsic origins of replication and are replicated by forks coming from adjacent early and late replicating domains [50,51].

## 3. Deregulation in Origin Licensing

### 3.1. Origin Under-Licensing

Origin licensing is a highly regulated process, and it is strictly restricted to the end of M and G1 phases. In order to ensure the completion of the replication even in the presence of low or mild replication stress conditions, a largely excessive number of origins is licensed. As a result, the majority (90%) of licensed origins remain unused and dormant and will be passively replicated [34,35]. However, when an ongoing replication fork encounters a barrier that leads to its delay or stalling, nearby dormant origins get activated to rescue and complete DNA replication. Here comes the importance of efficient licensing in G1 [52]. Reducing the number of licensed origins puts the cells at risk of having an insufficient number of rescuing origins to complete and safeguard replication. As a result, the cells struggle to finish replication leaving unreplicated regions behind. Indeed, it frequently happens at genomic regions characterized by a low level of origins, as the common fragile sites (CFS), known under-licensed regions of the genome. They represent hotspots of DNA breaks and chromosomal rearrangements [53,54].

Accordingly, a reduced level of the pre-RC component MCM5 led to a decrease in the loading of complete MCM complexes, reducing the number of licensed and potentially active origins leading to DNA damage in response to DNA polymerases inhibitors aphidicolin (APH) or hydroxyurea (HU) [34]. A similar outcome was also reported by Ekholm-Reed and collaborators in cells overexpressing cyclin E which led to premature S-phase entry resulting in reduced levels of loaded MCM4 and MCM7 [43]. Since origin licensing in G1 occurs progressively [55], a premature S-phase entry leads, de facto, to a reduction of the number of licensed and dormant origins [7,8]. This situation leads to replication stress and genomic instability causing human pathology.

### 3.2. Unscheduled (Anticipated) Origin Licensing

Macheret and colleagues, by analyzing the consequence of Cyclin E overexpression on cell behavior, were the first to introduce the concept of unscheduled or ectopic origin licensing. Since origin licensing occurs in G1, intragenic pre-RCs are usually passively displaced outside the intragenic regions of the actively transcribed genes by ongoing transcription machineries during the G1. Thus, physiologically, fired origins are localized near the gene promoter outside the transcribed body of the gene. Cyclin E overexpression leads to a premature entry of the cell in S-phase, somehow ‘forcing’ the pre-RCs to be fired before their displacement. According to origin mapping data, a new landscape of fired origins can be observed in cells overexpressing cyclin E compared to the control. The firing of the intra-genic origin while transcription is underway exposes the replication forks encountering the transcription machinery to the risk of delay, stall or collapse which could explain the genome instability observed in cyclin E-overexpressing cells [5]. Cyclin E was shown to be deregulated in several human cancer which demonstrated high aggressiveness, poor prognosis, and resistance to chemotherapy [56,57,58,59,60,61,62]. Another protein that positively controls the G1/S transition is the cyclin D1 and, similarly to cyclin E, its overexpression affects S-phase entry and origin licensing. Cyclin D1 overexpression constitutes an indicator of poor prognosis in non-small lung carcinoma [63]. Similar results were also seen with *MYC* overexpression [5]. MYC-driven oncogenesis has also been correlated with poor prognosis in breast [64] and in pancreatic ductal cancer [65].

### 3.3. Origin Re-Licensing

An unrestrained, erroneous reassembly and licensing of the origins immediately after the duplication of a chromosome region can cause its re-replication during the current S-phase. This results in having additional copies of the same region. Almost all of the studies addressing origin re-licensing and DNA re-replication were carried on in cells overexpressing licensing proteins including CDT1 and CDC6. In 2017, Munoz and colleagues showed that Cdt1 overexpression induced re-replication and tissue dysplasia in vivo [66]. The same study showed that the deregulation of Cdt1 or Cdc6 expression led to an increase in p21, a marker of DNA damage response. The failure of activating this pathway in cancer might explain how these cells survive despite the occurrence of re-replication [67,68,69,70]. However, whether de novo re-replication really occurs in physiological settings is still an unresolved question.

## 4. Replication Fork Progression

Once fired, the route of a replication fork does not resemble the descent of a peaceful river but rather a jumping competition. A replication fork will encounter many difficulties and barriers that will oppose its advancement.

### 4.1. Alteration in the Nucleotides Pool

The first difficulty for an ongoing fork is to have sufficient fuel to progress. Indeed, to guarantee a normal progression of the replication forks and the on-time completion of the S-phase DNA polymerases need an equilibrated intake of deoxyribonucleotides to sustain the polymerization of the new DNA strands. Thus, deoxyribonucleotides levels and relative ratios are important to ensure the normal progression, i.e., the speed, of a replication fork. Predictably, alterations in the available nucleotide pools reduce fork velocity and force the cell to fire dormant origins to compensate, which, in turn, will aggravate the lack of nucleotides by causing fork stalling or leading to a ‘complete’ S-phase transit with many under-replicated zones that will be eventually duplicated later or will lead to the formation of DNA breaks.

Accordingly, the role of unbalanced nucleotide pools in replication stress and genomic instability has been stressed by treating cells with hydroxyurea, a ribonucleotide reductase inhibitor that alters nucleotides biogenesis associating fork delay or arrest and DNA double-strand breaks (DSBs) accumulation as consequence of stalled replication forks [71,72,73,74].

### 4.2. Barriers to Fork Progression

#### 4.2.1. G-Quadruplexes

Non-B secondary structures, largely determined by the sequence context, as in the case of palindromes, can lead to alternative DNA tridimensional folding influencing the progression of the replication fork per se. Beyond palindromes and associated hairpins, a G-rich repetitive sequence can fold on itself forming a four-stranded intramolecular structure, called a G-quadruplex [75]. G-rich regions were mapped upstream of transcription start sites (TSSs) and the G4 structures are enriched in actively transcribed genes, either to favors the access of transcriptional machineries to TSS or as a passive consequence of the DNA opening due to transcriptional machinery access at TSS. G4s located on the non-template strand may also facilitate the progression of the transcriptional machinery on the template strand of highly transcribed genes, such as the DNA sequence encoding the rRNAs. Whatever the reasons of their enriched presence at TSS of transcribed genes, G4s are thought to facilitate the access of the replication machinery at replication origins located in the proximity of TSS. This also explains the enrichment of early replication initiation events at transcriptionally active sites [76,77]. Inversely, it has also been shown that the nuclear level of G4s increases significantly during S phase [78], indicating that G4s positively associate with the opening of the DNA double-helix, again either to allow replication machinery access to the DNA template or as a passive consequence of the fork progression. In addition, while a DNA polymerase encountering a DNA lesion slows down or halts its progression, the replicative helicase can continue to unwind the double helix leaving behind a long stretch of single-stranded DNA (ssDNA). This ssDNA could be at high risk of G4s formation, constituting an additional barrier for the already delayed DNA polymerase. Indeed, for their relative stability that is potentially increased by their interaction with specific proteins, G4s could also act as a barrier to fork progression and a threat for genomic stability. Indeed, G4s have been shown to be enriched at DSB, deletion and translocation hotspots, suggesting a possible role for G4s in inducing genome instability [1], as supported by initial observations in DT-40 cells, where it has been demonstrated that the presence of G4 motifs at TSS exacerbates the hydroxyurea-induced genomic instability at the BU-1 locus [79].

Thus, it appears that G4s have a dual role in DNA replication: on one hand, by ‘opening’ or ‘relaxing’ the DNA at replication origins, they facilitate replication initiation; on the other hand, their presence, promoted by the opening of the double helix during replication fork progression, can act as a major obstacle for ongoing forks. In parallel with the formation of G4 structures on one DNA strand, an i-motif forms on the complementary cytosine-rich strand. Whether G4s formation is dependent on i-motif formation or vice versa is controversial suggesting their interdependence [80].

#### 4.2.2. Replication–Transcription Conflicts

The replication and transcription machineries travel on the same route, in the same or the opposite direction. As a result, they risk meeting head-to-head or head-to-back. Since they are “copying” the template in the same 5′-to-3′ direction and at comparable speeds, 1–3 kb/min and 0.5–5 kb/min on average for replication and transcription, respectively, the head-to-back accident is a rare event but favored by the slowdown of the engine ahead. In contrast, head-to-head convergence between the RNA and DNA polymerase machineries copying opposite strands of the same DNA sequence is a more common occurrence also under physiologic, unstressed, conditions. Such encounters can delay the progress of both machineries due to the accumulation of supercoiled DNA in the space between the two.

A nascent RNA template can hybridize behind the RNA polymerase machinery with the copied DNA strand forming an RNA:DNA hybrid called R-loop. Physiologically, R-loops play a role in mitochondrial replication [81], transcription initiation and termination [82,83,84,85,86,87,88], and DNA repair [89,90,91]. However, the R-loop intrinsic stability increases as a consequence of the delay imposed at the progression of transcriptional machineries encountering a replication fork. Such stabler R-loops, representing an additional threat to replication progression, may become a threat for genome stability. Notably, some long genes, often hosted at CFS, characterized by a transcription that extends beyond a complete cell cycle [2], undergo increased breakage only when actively transcribed, suggesting that transcription is a co-factor responsible for CFS expression, i.e., breakage. Supporting the ‘negative’ role of R-loops in such fragility, DNA instability can be largely rescued upon the ectopic overexpression of the RNase H1 which reduces RNA:DNA hybrids [2]. Thus, the level of R-loops is highly regulated in mammalian cells to prevent genome instability. Beyond Rnase H1, several other proteins are involved in such regulation, including topoisomerases, RNA processing factors, and chromatin modulators [92] as well as the protein Senataxin and Aquarius, which act as helicases unwinding the RNA:DNA hybrids [85,93]. Inversely, loss of R-loop suppressors leads to the accumulation of RNA:DNA hybrids that are commonly associated with genome instability. How can R-loops, behind a transcription apparatus represent a supplemental obstacle to the progression of this last? One possibility might be that the presence of R-loops on DNA template creates a supplemental torsional stress ahead of the ongoing replication fork on the lagging strand. This torsional stress might affect fork progression leading to fork arrest. Another possibility might be that the physical presence of RNA on the DNA template might interfere with RNA polymerase that primes Okazaki fragments on the lagging strand affecting fork progression.

However, whether R-loops are a cause or a consequence of fork blockage is still debated. Since most studies have shown that overexpression of RNase H1, disrupting RNA:DNA hybrid, restores fork speed, it has been proposed that R-loops per se lead to replication fork delay/blockage. On the contrary, fork stalling due to replication stress would leave the downstream ssDNA exposed favoring the formation of RNA:DNA hybrids. Indeed, analysis on post-replicative repair mutants or mutant cells with the loss of DNA damage checkpoint have been associated with an increase in RNA:DNA hybrids due to persistent fork stalling. [94,95]. The different physical inducers of replications stress during different phases of the cell cycle are summarized in (Figure 2).

#### 4.2.3. DNA Damage

Whatever their origin and chemistry, DNA lesions associated or not with a modified chromatin structure, are major barriers for replication fork progression and a driver of replication stress. Encountering a not previously repaired ssDNA break or a DSB will cause the derailment of the replication machinery and the conversion of the primary lesions in an extremely cytotoxic locally altered DNA structures (see Figure 3A). The collision of the canonical replication machinery with a chemically modified base on one of the opposite strand results in the partial stall of the DNA synthesis on the damaged strand, but not on the opposite one, possibly generating a long stretch of ssDNA (see Figure 3B). Finally, the presence of lesions linking both strands leads to a full block of the replication fork (see Figure 3C). These damages are caused by several endogenous and exogenous factors that have been previously reviewed [11].

To avoid replication stress associated to DNA damage and/or limit its consequences, the cells possess three alternative strategies: either eliminate the specific lesion by using one of the well described dedicated DNA repair pathways [96] before the arrival of the replication forks; use a more permissive DNA polymerase or homologous recombination-associated mechanisms to by-pass the lesions [97]; or ‘eliminate’ the stalled replication machinery and protect/stabilize the fork allowing the removal of the barrier and the resuming of replication by HR-associated mechanisms [98]. The PRIMase and DNA-directed POLymerase (PRIMPOL) is an additional player involved in tolerating DNA damage [99,100]. Possessing both primase and polymerase activity, PRIMPOL initiates repriming downstream the DNA lesion, through interaction with RPA-bound ssDNA [101,102,103], on the leading strand with preferential incorporation of deoxyribonucleotides triphosphate over ribonucleotides [99,100,101]. These primers can be elongated by the replicative polymerase ε [99]. More recently, Gonzalez-Acosta and colleagues highlighted on the role of repriming of PRIMPOL in traversing DNA Intra-strand Cross Links (ICL). The mechanism by which PRIMPOL is recruited was recently shown to be independent of FANC/BRCA pathway [104] indicating that these proteins act in separate pathways in promoting ICL traverse.

## 5. Mechanisms for Managing Replication Stress

An integral part of the DDR network, the proteins embedded in the FANC/BRCA pathway as well as their activators, companions and targets constitute the main force directly or indirectly involved in the elimination of some replication barriers, including G4s, R-loops, and DNA inter-strands crosslinks at the stalled fork. Moreover, the FANC/BRCA pathway is involved in replication fork protection and in its rescuing, ensuring the S-phase progression as well as in mediating G2/M events allowing mitosis completion and chromosome segregation, limiting genetic instability in the daughter cells. Even if its loss-of-function leads to a cellular and chromosomal hypersensitivity to DNA Inter-strands Cross Links (ICL) [20,105] but at a minor extent to replication inhibitors [106], the FANC/BRCA pathway is activated in every situation of replication stress. Accordingly, the FANCcore complex-mediated monoubiquitination of FANCD2 and FANCI, readout of the pathway activation, has been observed in response to each kind of genotoxic stress, including UV [107] and ionizing radiation [108]. However, its activity is absolutely required only to cope with replication stress associated to the ‘simultaneous’ stall of both the replication machineries working on the leading and lagging strand of the same fork, as is the case in response to the replication inhibitors APH or HU or when fork meets an ICL [18]. Beyond being induced by antiproliferative agents as mitomycin C, cisplatin, or photoactivated psoralens, ICLs originate spontaneously as consequences of the oxidation of certain aldehydes, product of the cellular metabolism [109,110,111,112,113,114]. In the other situations, the FANC pathway represents a back-up solution that intervenes when the other strategies, HR-mediated by-pass [115,116] or TransLesional Synthesis (TLS) synthesis [117], are overwhelmed.

### 5.1. FANC/BRCA Pathway-Mediated Replication Rescue during S Phase

Responding, respectively, to replication fork delay and ssDNA accumulation, FANCM and RPA act as replication stress sensors, participating in the full activation of the main signaling pathway involved in fork rescue during S phase: the ataxia telangiectasia and Rad3-related (ATR) and checkpoint kinase1 (CHK1) axis, which in turn phosphorylate FANCM and RPA to amplify the stress signal and activate several downstream mediators and effectors directly involved in replication rescuing [118,119,120,121,122,123].

A main function of the ATR/CHK1 pathway is the activation of local dormant origins to rescue replication and/or to the inhibition of later origin clusters. This ensures the completion of replication at the site of the arrested forks and prevents the exhaustion of required factors. Thus, ATR/CHK1 pathway acts as a key regulator of the S-phase progression [124], during unperturbed cell cycle [125,126]. Loss of CHK1, in absence of other stress, is sufficient to lead to accumulation of chromosomal and mitotic abnormality [127,128]. Activated CHK1, by phosphorylating CDC25 inactivates CDKs, whose activity is required for both replication origin firing and timely progression of the replication timing. Consequently, CDKs inactivation delays S-phase progression [129].

A stalled fork can collapse, generating a one-ended DSB, leading to the activation of the ataxia telangiectasia mutated (ATM) and checkpoint kinase 2 (CHK2) pathway downstream the assembling of the MRE11-RAD50-Nibrin (MRN) complex, a major DSB sensor. Physiologically, as the ATR-CHK1 signaling pathway, the ATM-CHK2 activity is required to activate and coordinate pathways involved in cell cycle regulation and DNA repair. As the ATR-CHK1 axis, ATM and CHK2 phosphorylate several downstream substrates some in common with ATR, including members of the FANC/BRCA pathway [130,131].

Whereas a classical two-ended DSB will be repaired by the pre-replicative canonical non homologous end joining (cNHEJ) or by the post-replicative homologous recombination pathways or one of their several sub-pathways [132], the rescue of a one-ended DSB associated to a stalled fork requires the mandatory use of a HR sub-pathway called Break-Induced Repair (BIR). Otherwise, the unscheduled usage of the NHEJ pathway in this setting will lead to the formation of gross chromosomal abnormalities, including deletions and chromosome fusions as observed in FA [133,134,135]. The key steps of the BIR process involves the resection of the DSB extremity possibly involving MRN complex [136]. This creates a 3′-ssDNA stretch which will be first covered by RPA that will be displaced by BRCA2 to allow the formation of the ssDNA-RAD51 filament [137].

A successive D-Loop is formed and DNA polymerases are recruited to restart DNA synthesis. Importantly, fork stabilization, breakage, and rescue via the BIR is largely coordinated and executed by proteins of the FANC pathway. The chromatin-associated foci formed by the monoUbFANCD2/FANCI represent the hub and the link between the sensing of the stalled forks and the ‘enzymatic’ events that lead to its rescuing. The foci coordinate the come-and-go and the activities of several endonucleases, including FAN1, MUS81-Eme2, SLX1-SLX4, and XPF-ERCC1 [12,138] and TLS polymerases, such as Pol η, Pol θ, and Rev1-Rev7 [107,117,139], to eliminate or bypass the replication barrier, if any, and induce the DNA one-ended DSB. The foci also participate in the protection and stability of the local structure and allow the assembly of the MRN complex and the HR-mediated replication restart [136].

In addition to the previous role, components of the FANC pathway as FANCD2 and, mainly, the helicase FANCJ as well as other FANC pathway functionally associated helicases, including Bloom (BLM) and Werner (WRN) helicases, participate in the replication stress response by unwinding G4s and, expectedly, their loss-of-function leads to increased genetic instability and chromosome fragility [140,141,142,143].

### 5.2. Replication Rescue beyond S Phase

#### 5.2.1. During G2/Mitosis

Although endowed with many systems to cope with replication difficulties, replication can fail to be completed inside the ‘canonical’ S-phase boundaries and the cell can progress in G2/M despite the presence of DNA regions not being fully replicated. Indeed, under replication stress condition, the activation of S-phase checkpoint prevents mitotic entry to ensure proper duplication of the genome. However, it has been shown that cells depleted of the licensing factor Cdc6 undergo mitotic division in the absence of DNA replication [144]. This implies that S-phase checkpoints are only activated if forks were stalled or collapsed. Thus, despite having under-replicated regions in S phase, cells would still undergo mitosis.

Under-replicated regions in unstressed conditions typically happen at CFS, late-replicating regions poorly equipped in licensed replication origins [53,54]. Under low/mild replication stress, CFS regions, having insufficient dormant backup origins to awake to complete replication coupled to a limited time frame before reach G2, can enter G2 and progress to M with under- or mis-replicated, i.e., entangled, regions. This leads to chromosomal breaks seen on metaphase chromosomes also in absence of a genotoxic treatment. Although representing an obvious biological threat for the genome integrity, such events are also important for maintaining genome flexibility. CFS are conserved throughout evolution and colocalize with evolutionary breakpoints [145]. However, even if CFS breakage participates to the evolutionary processes, several mechanisms have evolved beyond S-phase repair to complete CFS replication, in order to avoid the threat of their breakage on the genome stability of the somatic cells. Several assays have been used to assess replication stress consequences in mitosis and in the following cell cycle; reviewed in [146].

FANC pathway has been also involved in protecting under-replicated regions from S phase till mitosis (see Figure 4) [147,148,149,150]. Seminal work of the Hickson and Rosselli groups demonstrated that G2 and mitotic cells progressed from S-phase present FANCD2/FANCI foci, the frequency of which is increased under exogenous replication stress. Notably, each FANCD2 focus is constituted by ‘twin’ spots [149,151]. When unresolved, such twin foci symmetrically segregate during ana-telophase upon sister-chromatid separation, demonstrating that each of them assembles on and protects each chromatid of a chromosome. Importantly, the mitotic FANCD2 foci represent S-phase unresolved foci that transit through G2 and M rather than the formation of monoUbFANCD2 on lesions induced in G2/M cells. Indeed, no foci are observed in cells that were in G2 at the moment of the genotoxic treatment. On the mitotic chromosome, during pro- and metaphase, the twin FANCD2 spots colocalize with a single spot of the endonucleases MUS81-Eme2, SLX1-SLX4, and XPF-ERCC1 located in between them [12]. Later, at ana-telophase, where henceforth endonucleases-free unresolved regions leads to the formation of ana-telophase bridges, FANCD2 foci segregate and participate in the optimal assembling of BLM on the DNA bridge to resolve it limiting chromosomal mis-segregation and DNA breakages [149]. Validating that late replicating regions with replication problems are the APH-induced FANCD2 foci in G2/M that co-localize with CFS [151].

The FANCD2 twin spots on G2 and M cells were also shown to colocalize with EdU foci detected by immunofluorescence after the incorporation of the nucleotide analog following a short pulse of 20–30 min [12]. Thus, the persistence of FANCD2 foci in mitosis marks unresolved DNA intermediates and their co-localization attest that the cells continue to synthesize DNA beyond S-phase hoping to complete the replication before cytokinesis. An issue of such situation is that, in the same cell, it is possible to observe EdU and FANCD2 co-localizing foci as well as EdU and FANCD2 sole foci (see Figure 5A). The presence of EdU foci in the absence of FANCD2 might be explained by either the displacement of FANCD2 after efficient resolving of that particular site or by the existence of a FANCD2-independent DNA repair pathway in G2/M. On the other hand, the presence of FANCD2 foci in the absence of EdU suggests that either the site has not been repaired yet or that, definitively, it cannot be repaired (see Figure 5B), though it is worth mentioning that most of EdU foci colocalize with FANCD2 in APH-treated cells (see Figure 5A), implying an abortive or an incomplete trial of the cell to locally achieve DNA synthesis.

Later work showed that this EdU incorporation is POLD3-dependent and is referred to as mitotic DNA synthesis (MiDAS). Depletion of MUS81 resulted in reduced recruitment of POLD3 suggesting the role of MUS81 in POLD3 recruitment [14]. EdU foci formation can be attributed to two not mutually exclusive reasons: either an extreme delayed attempt to complete DNA replication before mitosis or cytokinesis, or represent the consequence of a DNA reparative synthesis, a kind of chromosome bandage upon an injury. All these mechanisms ensure proper chromosomal segregation in mitosis.

#### 5.2.2. G1 of the Next Cell Cycle

Despite the involvement of several mechanisms to rescue under-replicated regions, some of them might escape the process. This causes ana-telophase bridges, leading to chromosome mis-segregation or DNA breakage in the successive G1 cells. In the next G1, these DNA lesions are bound by 53BP1 or RPA to protect them until the onset of S phase. The 53BP1 nuclear bodies [152] then dissolve during the successive S phase indicating that active replication of these lesions is taking place. RAD52 was also shown to be associated with these regions. As a result, the frequency of ultrafine bridges was shown to be reduced in the second cell cycle compared to the first [153]. This indicates that DNA repair during G2/M might not be sufficient to rescue under-replicated regions. As a result, the cells protect these regions with nuclear bodies to be able to repair them using active replication during S phase.

Another mechanism by which cells protect post-mitotic lesions is through RPA binding. RPA was shown to form foci in G1 cells exposed to APH in the previous S phase. However, immunofluorescence experiments showed that RPA staining pattern was different than 53BP1 nuclear bodies one, suggesting that these proteins associate and protect different genomic lesions. These RPA foci colocalize with single stranded DNA that escaped the previous cell cycle, and they predominantly associate with telomeric regions. Knocking down RAD52 reduced G1-associated RPA foci indicating that these regions might have been primed by RAD52 in the previous mitosis [154].

#### 5.2.3. Mitotic DNA Synthesis: A Fact or an Artifact

Later work from Hickson laboratory emphasized the involvement of MUS81, RAD52 and POLD3 in performing MiDAS [13]. Notably, about 50% of the sequenced EdU-containing regions colocalized with known CFS. The other 50% are probably non-identified CFS or not CFS [155]. Indeed, RAD51 recombinase was also shown to be essential for MiDAS to occur. Depletion of RAD51 in U2OS resulted in reduced mitotic EdU incorporation associated with reduced mitotic population and increase γ-H2AX foci. This implies that MiDAS is a mechanism to rescue DNA integrity before chromosome segregation during anaphase. The onset of anaphase was shown to be delayed when MiDAS is inhibited [156].

Recently, it has been reported that the frequency of MiDAS is elevated in BRCA2 deficient cells [157]. However, most of the EdU-sequenced regions in BRCA2-deficient mitotic cells did not overlap with APH-induced MiDAS. Instead, MiDAS triggered by BRCA2 loss colocalized with R-loops. According to the authors, the overlap of MiDAS with R-loops might result from unresolved transcription–replication conflicts in the preceding S phase. These DNA lesions turnout to be the legacy of DNA repair in mitosis. This phenotype was restored upon overexpression of RNase H1 highlighting on the R-loops as a possible inducer of MiDAS [157].

Thus, it appears that the occurrence of MiDAS is dependent on several proteins mainly linked to the HR pathway, including RAD51- and RAD52-dependent mechanisms. RAD52 pathway might be used to rescue CFS whereas RAD51 pathway might be employed to rescue other under-replicated regions. In any case, these studies emphasize the importance of MiDAS in rescuing under-replicated DNA lesions before the onset of anaphase. This might be the last salvage mechanism before chromosome mis-segregation.

In contrast to the previous posit, more recently, Mocanu and colleagues showed that CDK1 inhibition in CDK1-truncated mutants did not present with the phenotype seen using the CDK1 inhibitor RO3306 [158]. Their results showed that in cells treated with RO3306, but not in CDK1-mutant cells, EdU incorporation persisted in late G2 cells as well as in mitosis indicating that RO3306 per se compromises DNA synthesis. They also showed that depletion of MUS81 or POLD3 did not reduce EdU incorporation in mitotic cells of CDK1-truncated mutants, in contrast to what was shown by Hickson lab using RO3306 CDK1-inhibitor. On the basis of their observations, they claim that the EdU incorporation observed in mitotic cells is a DNA synthesis run over from the previous S phase rather than an actual mitotic DNA synthesis and that the previous data are due to an undetermined not specific CDK1-independent actions of the RO3306 inhibitor. This might have altered S-phase dynamics and created some DNA structures that needed to be resolved by MUS81 and POLD3 in mitosis. According to the Mocanu and colleagues hypothesis, the cells might enter mitosis with persistent ongoing forks that continue DNA synthesis in M. However, the mechanism by which DNA synthesis is resumed is probably through an alternative pathway that differs from the canonical S-phase replication. Indeed, they also showed that depletion of RAD51 or RAD52 reduced EdU incorporation in both G2 and mitotic cells, similar to data from the Hickson lab. Altogether, the results indicate that the EdU incorporation on mitotic cells is either a DNA repair mechanism or a continuation of replication of under-replicated regions as we have previously suggested [12,149]. On the other hand, we should not ignore that chromosome condensation per se might induce breakages in some under-replicated regions. This might exacerbate DNA damage requiring the intervention of some kind of mitotic DNA repair. Thus, the term “Mitotic DNA synthesis” is probably to be revisited to reflect more precisely the nature of the events associated with the EdU spots observed on the mitotic chromosomes. Finally, some unresolved questions remain in addressing the actual necessity of FANCD2 in resuming DNA synthesis in G2/M and the existence of alternative pathways that might be involved in rescuing some type of replication abnormalities.

## Figures and Tables

**Figure 1 ijms-23-11331-f001:**
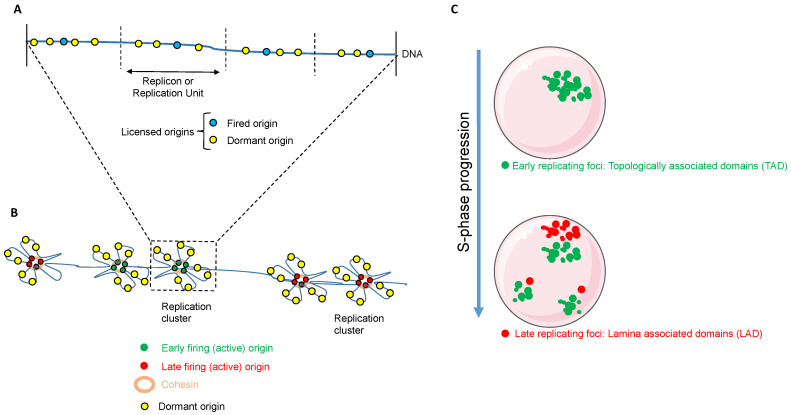
Spatiotemporal regulation of DNA replication. (**A**) A replicon represents the basic unit (50–450 kb), defined as a DNA segment replicated by a single active origin (blue). (**B**) Adjacent replication units are tethered together by cohesins to form a replication cluster. Origins from the same replication cluster fire at the same time either early (green) or late (red). Several replication clusters fired at the same time form a replication domain. (**C**) Early replicating foci (≈ replication domains) are located towards the anterior of the nucleus (green). They are replicated at the beginning of S phase and organized as TADs. Late replicating foci are located more towards the nuclear lamina hence organized as LADs (red) and are replicated at the end of S phase.

**Figure 2 ijms-23-11331-f002:**
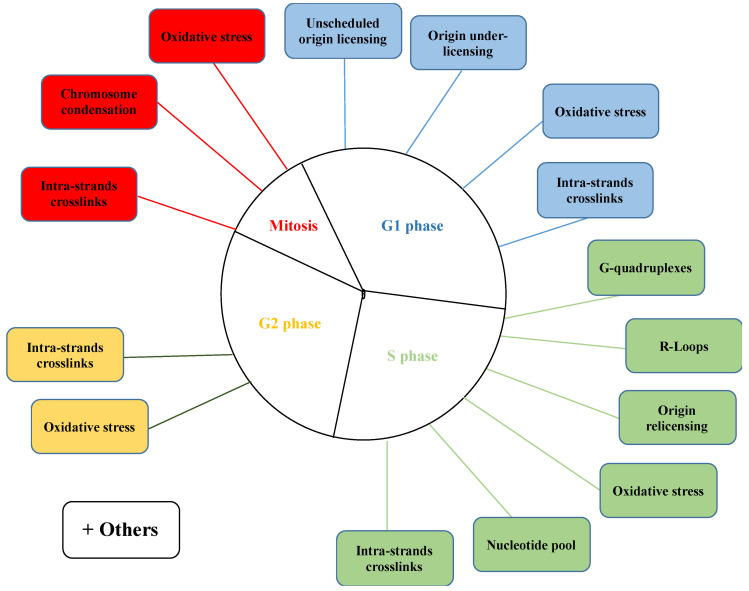
Inducers of replication stress across different phases of the cell cycle. This scheme summarizes the possible obstacles during G1 (blue), S (green), G2 (yellow), and mitosis (red) that affect fork progression, thus preventing efficient DNA replication.

**Figure 3 ijms-23-11331-f003:**
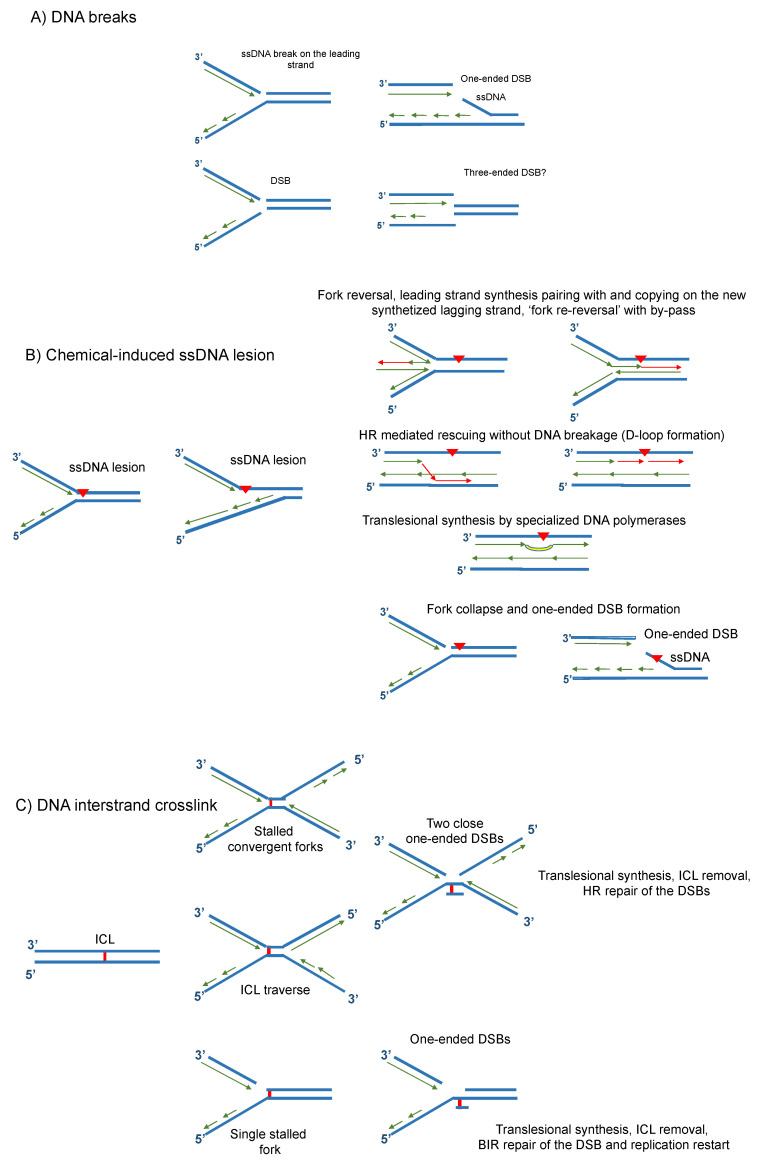
Possible outcomes of replication forks meeting a DNA lesion. (**A**) The replication fork might encounter an unrepaired ssDNA break on the leading strand or a DSB which will create a one-ended or a three-ended DSB. (**B**) The fork might also encounter chemically modified bases affecting one strand. Halted replication in this case can be rescued using several mechanisms including fork reversal, fork re-reversal with bypass, HR-mediated D-loop formation, and translesional synthesis to prevent fork collapse and associated DNA breakage. (**C**) An ICL, linking the two opposite strands of the same DNA molecule, can also block fork progression. Two convergent forks or a single fork can be affected by the recently described ICL traverse mechanism, that allows replication progression, leaving the lesion and a short unreplicated sequence behind the fork. Several repair proteins are activated to remove the ICL, repair the associated DNA repair-induced DSB, and rescue replication through HR-mediated mechanisms.

**Figure 4 ijms-23-11331-f004:**
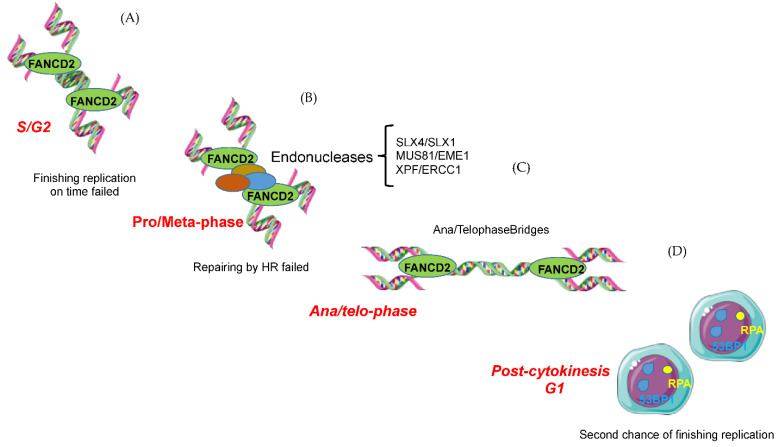
The role of FANCD2 in rescuing under-replicated regions throughout the cell cycle. (**A**) Upon replication stress during S phase, FANCD2 is recruited to the chromatin, as twin foci, to activate downstream repair pathways. (**B**) However, cells can still enter mitosis with under-replicated regions. Thus, to cleave under-replicated/untangled DNA, endonucleases are recruited between the chromatin-bound FANCD2 foci. (**C**) If left uncut/unrepaired, under-replicated chromatids will remain entangled forming ana/telophase bridges during chromosome segregation. (**D**) Unresolved anomalies in under-replicated regions covered by FANCD2 will be covered by RPA or 53BP1 to be protected in the next G1 and actively replicated in the next S phase.

**Figure 5 ijms-23-11331-f005:**
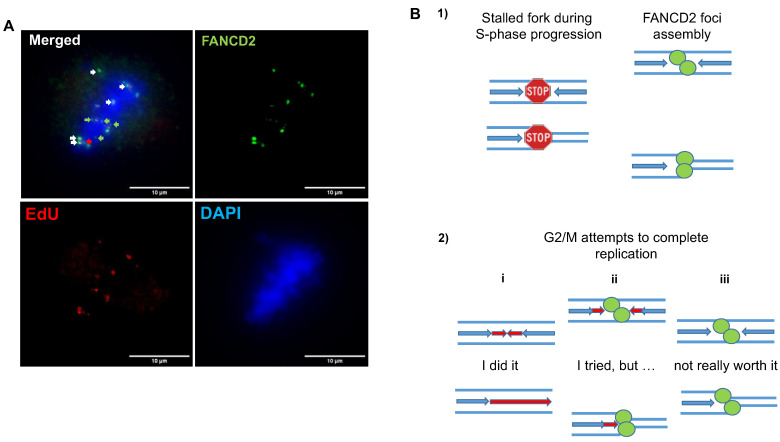
Colocalization of FANCD2 foci with EdU on mitotic cells. Colocalization of FANCD2 foci with EdU spots on mitotic cells. (**A**) APH-treated HeLa cells display FANCD2 foci (in green) that extensively colocalize (white arrow) with EdU spots (red). However, in some cases, FANCD2 foci (green arrows) as well as EdU spots (red arrow) exist as isolated signals. (**B**) (1) During replication stress, a single or two convergent forks are blocked, inducing the recruitment of FANCD2 twin foci to chromatin on both DNA templates (green foci) in S phase. (2) (i) During the next G2/M, under-replicated FANCD2-related regions synthesize DNA (red arrows) completing DNA replication and FANCD2 foci are eliminated, explaining the presence of single EdU foci. (ii) In the second scenario, replication cannot be totally completed. Thsu, FANCD2 foci persist colocalizing with EdU spots (red arrows and green foci). (iii) The third scenario is based on the persistence of FANCD2 foci in the absence of EdU involving an aborted DNA synthesis in G2/M (scale bar = 10 µm).

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
