# Peer review of "Safeguarding DNA Replication: A Golden Touch of MiDAS and Other Mechanisms"

_ijms, 2022, doi:10.3390/ijms231911331_

Round 1
Reviewer 1 Report
The review is easy to read however I had problems with the structure of the text.
For exemple B) Mechanisms for Managing line 310. Where is A?
Is FANC/BRCA Pathway Mediated Replication... line 332 a title? I yes should it be C)?
The focus of the review represent a key question in the field of genome stability and replication defects. The review includes several recent papers from 2022, and finishes in an interesting and challenging analysis supported by strong published results.
requested changes:
References would appreciated at line 40,126,319,331
extra space need to be deleted line 268 "over expression of the RNase"
Prim Pol is not mentioned in this review but in my opinion it is one of the main lesion bypass mechanism and it should be discuss in the review.
2 papers on Replication/transcription replication stress and Midas has been accepted this week in Molecular cell. Introducing these papers in the review will significative increase its impact
Author Response
First, we would like to thank the reviewer for the positive evaluation of our manuscript.
The review is easy to read however I had problems with the structure of the text.
For exemple B) Mechanisms for Managing line 310. Where is A?
Is FANC/BRCA Pathway Mediated Replication... line 332 a title? I yes should it be C)?
We modified the structure of the text to be coherent and easy to follow.
The focus of the review represent a key question in the field of genome stability and replication defects. The review includes several recent papers from 2022, and finishes in an interesting and challenging analysis supported by strong published results.
Requested changes:
References would appreciated at line 40,126,319,331
As requested, additional references have been added to each line.
Extra space needs to be deleted line 268 "over expression of the RNase"
The text has been corrected.
Prim Pol is not mentioned in this review but in my opinion, it is one of the main lesion bypass mechanism and it should be discussed in the review.
We added a new paragraph concerning Prim Pol, line 335-345.
2 papers on Replication/transcription replication stress and Midas has been accepted this week in Molecular cell. Introducing these papers in the review will significative increase its impact
Thanks again for the suggestion, both articles are now cited and discussed.
Reviewer 2 Report
AL-AHMAD-NACHAR and ROSSELLI discussed the regulation of DNA replication and a variety of potential barriers for DNA replication. They then focused on the FANC/BRCA related proteins and their roles in managing the replication stress. Finally, they reviewed the so-called MiDAS and its role in rescuing DNA replication before cytokinesis. The authors did a nice job in summarizing the progress in DNA replication and replication stress response. Since MiDAS is in the title of this manuscript and some intriguing discoveries were made recently related to it, I recommend that the authors should discuss more on why MiDAS is a potential artifact. With the following revisions, I recommend its acceptance for publication.
Major points: discuss more on why MiDAS is a potential artifact.
Minor points:
Line-244: NOT carbon-rich; it should be “cytosine-rich”
Line-273: “R-loop” instead of “R-loops”
Line-441: overwhelmed
Line-427: “the” instead of “a”
Author Response
First, we would like to thank the reviewer for the positive evaluation of our manuscript.
AL-AHMAD-NACHAR and ROSSELLI discussed the regulation of DNA replication and a variety of potential barriers for DNA replication. They then focused on the FANC/BRCA related proteins and their roles in managing the replication stress. Finally, they reviewed the so-called MiDAS and its role in rescuing DNA replication before cytokinesis. The authors did a nice job in summarizing the progress in DNA replication and replication stress response. Since MiDAS is in the title of this manuscript and some intriguing discoveries were made recently related to it, I recommend that the authors should discuss more on why MiDAS is a potential artifact. With the following revisions, I recommend its acceptance for publication.
Major points: discuss more on why MiDAS is a potential artifact.
According to the request of the referee, the final part of the review was amplified, from line 515.
Minor points:
Line-244: NOT carbon-rich; it should be “cytosine-rich”
Line-273: “R-loop” instead of “R-loops”
Line-441: overwhelmed
Line-427: “the” instead of “a”
All points have been corrected